# Protective Effects of Taraxasterol against Deoxynivalenol-Induced Damage to Bovine Mammary Epithelial Cells

**DOI:** 10.3390/toxins14030211

**Published:** 2022-03-15

**Authors:** Junxiong Wang, Kexin Zheng, Yongcheng Jin, Yurong Fu, Rui Wang, Jing Zhang

**Affiliations:** State Key Laboratory for Zoonotic Diseases, Institute of Zoonosis Research, College of Animal Sciences, Jilin University, Ministry of Education, Changchun 130062, China; jxwang19@mails.jlu.edu.cn (J.W.); zhengkx528@163.com (K.Z.); ycjin@jlu.edu.cn (Y.J.); fuyr20@mails.jlu.edu.cn (Y.F.); ruiwang18@mails.jlu.edu.cn (R.W.)

**Keywords:** taraxasterol, deoxynivalenol, endoplasmic reticulum stress, apoptosis, bovine mammary epithelial cells

## Abstract

Deoxynivalenol (DON), a mycotoxin produced by *Fusarium graminearum*, is one of the most prevalent contaminants in livestock feed and causes very large losses to animal husbandry every year. Taraxasterol, isolated from *Taraxacum officinale*, has anti-inflammatory, antioxidative stress, and antitumor effects. In the present study, bovine mammary epithelial cells (MAC-T) were used as a model, and different concentrations of taraxasterol (0, 1, 5, 10, and 20 μg/mL) were used to protect against DON-induced cell damage. The results showed that taraxasterol at a concentration of 10 μg/mL significantly increased cell viability. Analysis of lactate dehydrogenase (LDH) levels indicated that taraxasterol substantially decreased LDH release caused by DON. Taraxasterol effectively alleviated the depletion of glutathione (GSH), the increase in the lipid peroxidation of malondialdehyde (MDA), the reduction in total superoxide dismutase (T-SOD) activity, and the decrease in total antioxidant capacity (T-AOC) induced by DON. The results further showed that taraxasterol reduced the accumulation of reactive oxygen species (ROS). Taraxasterol was found to relieve endoplasmic reticulum (ER) stress by suppressing the expression of glucose-regulated protein 78 kDa (GRP78), activating transcription factor 6 (ATF6), activating transcription factor 4 (ATF4) and the transcription factor C/EBP homologous protein (CHOP), and reducing cell apoptosis by suppressing the expression of caspase-3 and Bcl2-associated X (BAX) and upregulating the expression of the antiapoptotic protein B-cell lymphoma-2 (Bcl-2). Our research results indicate that taraxasterol could alleviate DON-induced damage to MAC-T cells.

## 1. Introduction

Dairy products have long been advertised as excellent sources of nutrition that benefit gastrointestinal health and the immune system [1]. The dairy industry has become a crucial part of the modern economy. Deoxynivalenol (DON), also known as vomitoxin, is an epoxy-sesquiterpenoid that belongs to the type B trichothecenes. A mycotoxin mostly produced by *Fusarium graminearum*, DON is one of the most prevalent contaminants in livestock feed [2,3]. The dairy industry suffers significant losses due to residual DON every year [4]. Ingestion of DON remaining in the feed by dairy cows can lead to poor production performance and a decline in the quality of milk products [5]. DON is chemically stable during food processing and can enter the body through the food chain [6], eventually endangering human health. At the cellular level, DON has a range of detrimental effects on dairy cows by inducing oxidative stress in cells and cell apoptosis [7,8]. Previous studies in bovine mammary epithelial cells and HepG2 human hepatocellular carcinoma (HCC) cells have shown that one of the mechanisms by which mycotoxins induce cell damage is by triggering the endoplasmic reticulum (ER) stress response [9,10]. Therefore, whether DON damages cells using a similar mechanism should be explored, and an effective substance to prevent this damage is needed.

Taraxasterol, (3β,18α,18α)-Urs-20(30)-en-3-ol, is a pentacyclic triterpene that can be isolated from *Taraxacum officinale* F.H. Wigg [11], which is a traditional medicine with lactating, choleretic, diuretic, and anti-inflammatory activities. A study found that dairy cows treated orally with an herbal extract containing T. officinale showed an increase in milk yield [12]. As Chinese medicinal herbs have received more attention in recent years, taraxasterol has been proven to have multiple protective effects. Taraxasterol was reported to inhibit iNOS and COX-2 expression in LPS-stimulated RAW264.7 cells [13] and inhibit IL-1β-induced NO and PGE2 production in human chondrocytes in osteoarthritis, a chronic degenerative joint disease [14]. Previous studies have shown that taraxasterol inhibits proliferation and induces apoptosis in HepG2 and Huh7 HCC cells [15,16,17]. Taraxasterol also showed protective effects against ethanol-induced liver injury in mice by regulating the CYP2E1/Nrf2/HO-1 and NF-κB signaling pathways [18]. In vivo, taraxasterol was found to protect against LPS-induced acute lung injury and endotoxic shock in mice [19]. Taraxasterol has been proven to be an emerging protective agent with multiple positive effects. However, its protective effects against damage caused by mycotoxins have hardly been explored.

Hence, this study aimed to explore the protective mechanisms of taraxasterol against DON-induced damage in bovine mammary epithelial cells.

## 2. Results

### 2.1. Taraxasterol Improved the Decrease in Bovine Mammary Epithelial Cell Viability Induced by DON

Cell viability was detected by CCK-8 assay. When the concentration of DON was 0.2 μg/mL, the cell viability was significantly reduced (Figure 1A). Therefore, 0.2 μg/mL DON was selected for the experiments. Taraxasterol significantly improved cell viability when its concentration was less than 10 μg/mL, and its protective effect against DON-induced cell damage peaked at a concentration of 10 μg/mL (Figure 1B). Therefore, subsequent experiments were conducted using the combination of 0.2 μg/mL DON and 10 μg/mL taraxasterol.

### 2.2. Taraxasterol Alleviated the Cellular Damage Caused by DON by Reducing LDH Levels

The LDH content was detected to assess the integrity of the cell membrane. Greater LDH leakage indicated more severe damage to the cell membrane. After treatment with 0.2 μg/mL DON and 10 μg/mL taraxasterol for 24 h, the cell culture medium was collected to detect the LDH level using an LDH kit. LDH leakage was significantly higher in culture medium from MAC-T cells treated with DON compared with the control group and decreased after treatment with 10 μg/mL taraxasterol (Figure 2).

### 2.3. Taraxasterol Prevented Oxidative Stress Induced by DON

The protective effects of taraxasterol were preliminarily revealed by the results of the previous experiment. Its antioxidant capacity was further explored by detecting several antioxidant indices. After treatment with 0.2 μg/mL DON and 10 μg/mL taraxasterol for 24 h, the MDA content was significantly higher in the DON group than in the control group, while in two of the taraxasterol-treated groups, the MDA content was reduced to the same level as that in the control group (Figure 3A). GSH level was significantly reduced in the DON group compared with the control group, and the GSH level in the taraxasterol group was significantly higher than that in the DON group (Figure 3B). The T-AOC activity test indicated that after treatment with DON, the T-AOC activity in MAC-T cells tended to decrease, and in the DON + taraxasterol group, the T-AOC activity was significantly higher than that in the DON group. Meanwhile, the taraxasterol group showed a significant increase in T-AOC activity compared to that in the control group (Figure 3C). The T-SOD activity was significantly lower in the DON group than in the control group, and in the DON + taraxasterol group, the T-SOD activity was significantly higher than that in the DON group (Figure 3D). ROS fluorescence intensity was significantly higher in the DON group than in the control group, and the ROS fluorescence intensity of the two taraxasterol-treated groups decreased to the same level as that in the control group (Figure 3E,F). The detection of these oxidant indices proved that DON could induce oxidative stress in MAC-T cells and that taraxasterol could protect MAC-T cells against oxidative stress.

### 2.4. The Protective Effects of Taraxasterol against the DON-Induced Decrease in Mitochondrial Membrane Potential (ΔΨm, MMP)

After treatment with DON and taraxasterol for 24 h, the MAC-T cells were treated with methylbenzimidazole and the fluorescent probe dichloromethane iodide (JC-1) and incubated for 0.5 h. The MMP was significantly decreased in the DON group compared to the control group. Additionally, the MMP was significantly higher in the DON + taraxasterol group than in the DON group (Figure 4A,B). A decrease in MMP is a marker of cell apoptosis, so this result suggested that DON can not only cause oxidative stress but also eventually lead to cell apoptosis and that taraxasterol can reverse this process.

### 2.5. Taraxasterol Alleviated ER Stress Caused by DON

DON was shown to induce severe oxidative stress in MAC-T cells, and taraxasterol effectively protected against DON. Oxidative stress often leads to the accumulation of misfolded proteins in cells, thus inducing ER stress. To further verify the protective effects of taraxasterol, the relative expression of genes and proteins involved in the ER signaling pathway was detected by real-time quantitative PCR and Western blotting. The results showed that the gene expression of GRP78, ATF6, and CHOP were significantly upregulated in the DON group compared to the control group and significantly downregulated in the taraxasterol-treated groups compared to the DON group (Figure 5A). The GRP78, ATF6, and CHOP protein levels showed trends consistent with the regulation of mRNA expression, as they were significantly increased in the DON group compared to the control group and significantly decreased in the taraxasterol-treated groups compared to the DON group (Figure 5B). ATF4 plays an important role in ER stress and regulates the gene expression of CHOP; therefore, the protein level of ATF4 was also detected. As expected, the protein content of ATF4 was increased in the DON group compared to the control group and significantly decreased in the taraxasterol group compared to the DON group (Figure 5B). This indicated that DON induced severe ER stress in MAC-T cells and that taraxasterol could effectively inhibit this damage.

### 2.6. Taraxasterol Alleviated the Cell Apoptosis Induced by DON

ER stress is always followed by cell apoptosis, which was already preliminarily proven by previous experiment, as after treatment with DON, the MMP of MAC-T cells was significantly decreased. To further explore the induction of cell apoptosis by DON and the protective effects of taraxasterol, the relative levels of proteins involved in the apoptosis signaling pathway were detected by Western blotting. The relative protein levels of cleaved caspase-3/caspase-3 and BAX were significantly increased in the DON group compared to the control group and significantly decreased in the taraxasterol-treated groups compared to the DON group (Figure 6B,C). The relative protein level of Bcl-2 was significantly decreased in the DON group compared to the control group and significantly increased in the DON + taraxasterol group compared to the DON group (Figure 6D). The BAX/Bcl-2 ratio was significantly higher in the DON group than in the control group and significantly lower in the taraxasterol-treated groups than in the DON group (Figure 6E).

## 3. Discussion

DON intake by dairy cows is inevitable during the feeding process. Residual DON not only decreases the quality of dairy products but also endangers human health [20]. DON has been proven capable of inducing oxidative stress, the inflammatory response, and apoptosis in bovine mammary epithelial cells [21]. Additionally, the multiple protective effects of taraxasterol, a traditional Chinese herbal medicine, have been explored [11,19,22,23]. The goal of this research was to explore the underlying mechanisms of the protective effects of taraxasterol.

The cytotoxic effects of DON were first verified by treating MAC-T cells with different concentrations (0.05, 0.1, 0.2, 0.3, 0.5 μg/mL) of DON. The results showed that DON at a concentration of 0.2 μg/mL significantly reduced cell viability. MAC-T cells were simultaneously treated with both reagents to verify the protective effects of taraxasterol against DON-induced cell damage. Taraxasterol was found to be capable of promoting cell viability, which was reduced by DON. Similar effects were found in BV2 microglial cells [24]. At a concentration of 10 μg/mL, taraxasterol showed the strongest ability to protect against the decrease in cell viability induced by DON, and taraxasterol at this concentration also significantly increased cell viability. Thus, the combination of 0.2 μg/mL DON and 10 μg/mL taraxasterol was selected. The cytotoxicity of a compound can be measured by measuring the level of LDH released by cells [25]. Generally, an increase in LDH levels indicates increased cytotoxicity. The LDH level of the DON treatment group was significantly increased by DON treatment, but after treatment with taraxasterol, the LDH level was significantly decreased. To further explore DON-induced oxidative damage, the content of MDA, the activities of GSH, T-SOD, and T-AOC were tested. SOD is an antioxidant metalloenzyme present in organisms. It can catalyze the disproportionation of superoxide anion free radicals to produce oxygen and hydrogen peroxide, and SOD activity plays a vital role in the oxidant/antioxidant balance and is closely related to the occurrence and development of many diseases [26]. GSH can help maintain normal immune system function and has integrated antioxidant and detoxification effects [27]. Lower activities of SOD and GSH suggest a decline in the antioxidant capacity of cells. DON was found to be the leading cause of the decrease of SOD and GSH activities in cells in our study, which was also concluded by another study conducted by Zhang [28]. The results indicated that taraxasterol effectively alleviated the depletion of GSH, the increase in the lipid peroxidation of MDA, and the decrease of T-AOC activity caused by DON. To further verify the antioxidant capacity of taraxasterol, the intracellular ROS levels of MAC-T cells were detected. ROS are a byproduct of the normal mitochondrial metabolism and homeostasis [29]. An increase in intracellular ROS levels in cells indicates oxidative stresses. In our study, DON significantly increased the ROS levels of MAC-T cells, which implied more severe DON-induced damage to MAC-T cells. ROS are known to induce collapse of the MMP and eventually lead to apoptosis [30]. As expected, the MMP of MAC-T cells was significantly decreased after treatment with DON. In addition, the ROS levels in the taraxasterol-treated groups were significantly decreased compared to those in the DON group, which further revealed the potential mechanisms of the protective effects of taraxasterol. In previous studies, ROS have also been connected to ER stress [31], which is also a leading cause of apoptosis [32]. Thus, a subsequent experiment was performed to determine the connection between ER stress and the protective effects of taraxasterol.

The ER is a multifold membranous structure within eukaryotic cells that plays a major role in the synthesis of complex molecules required by the cell and the organism as a whole [33]. The ER is recognized as the primary site of the synthesis and folding of secreted, membrane-bound, and some organelle-targeted proteins [34]. The accumulation of unfolded or misfolded proteins in the ER leads to stress [35]. In our study, we focused on the GRP78-ATF6-CHOP signaling pathway to investigate the possible strategy by which taraxasterol protects against DON-induced cell damage. When ER stress begins, accumulated unfolded proteins in the ER bind GRP78 and thereby activate transmembrane receptors such as ATF6. This process explains the increased mRNA expression of GRP78 and ATF6 in MAC-T cells after treatment with DON. In addition, the mRNA expression of GRP78 and ATF6 in the taraxasterol treatment group was significantly suppressed. As CHOP can be modulated by ATF6, the expression of CHOP showed the same changes observed for ATF6 and GRP78. These changes explain how DON induces ER stress and the mechanisms by which taraxasterol protects cells against damage. However, if protein aggregation persists and stress cannot be resolved, prosurvival signaling switches to proapoptotic signaling [36]. ER stress is a complex biological process that is regulated by several signaling pathways. ATF4 is a member of another signaling pathway that promotes cell survival by inducing genes involved in amino acid metabolism, redox reactions, the stress response, and protein secretion [37]. However, not all the genes whose expressions are induced by ATF4 are antiapoptotic. The transcription factor CHOP, whose induction strongly depends on ATF4, is well known to promote apoptotic cell death. When DON induced the mRNA expression of ATF6 and ATF4, the expression of CHOP was promoted. Therefore, the expression of genes from the cell apoptosis signaling pathway was regulated. After treating with DON for 24 h, the protein level of Bcl-2 was significantly lower and the protein level of BAX was significantly higher than the control group. Taraxasterol was proven to be protective against these cellular changes in our study. By downregulating the protein level of BAX and upregulating the protein level of Bcl-2, the protein level of caspase-3 was also downregulated. Therefore, cellular apoptosis was alleviated. A similar mechanism was found by Liu [24], who researched the protective effects of taraxasterol against concanavalin A-induced acute hepatic injury in mice.

Taraxasterol is an extract from *Taraxacum officinale*, which is widespread in nature and exceedingly easy to obtain. Although taraxacum is a well-known traditional herbal remedy with a long history, until recently, only limited scientific information was available to justify the reputed uses [23]. Our research shows taraxasterol to be a good prospect for use in the dairy industrial field. Its multiple protective effects make it feasible to be a protective agent that can be added into the animals’ feed.

## 4. Conclusions

Taraxasterol was found to be an excellent protective agent against cell damage caused by DON. The decrease in cell viability, increase in LDH leakage and ROS levels, decrease in MMP, ER stress and apoptosis induced by DON were significantly alleviated by taraxasterol. However, the findings were all conducted at the cellular level, and the potential effects of the addition of taraxasterol to livestock feed remain to be further explored.

## 5. Materials and Methods

### 5.1. Chemicals and Reagents

DON (purity > 99%), hydrocortisone, penicillin–streptomycin, and insulin were purchased from Sigma–Aldrich (St. Louis, MO, USA). DON was dissolved in ethanol and stored at −20 °C. Taraxasterol was purchased from Refinsen Biotech Co., Ltd. (Chengdu, China), dissolved in dimethyl sulfoxide (DMSO, Sigma Chemical Co., St. Louis, MO, USA) and stored at 4 °C. Fetal bovine serum (FBS) was purchased from Gibco (Gaithersburg, MD, USA), and high-glucose Dulbecco’s modified Eagle’s medium (DMEM) was purchased from HyClone (Logan, UT, USA). The Cell Counting Kit-8 (CCK-8) was procured from Dojindo Laboratories (Kumamoto, Japan). Kits to measure ROS were obtained from Beyotime Biotechnology (Shanghai, China). Kits for detecting T-SOD, MDA, GSH, T-AOC and lactate dehydrogenase (LDH) activities were purchased from Jiancheng Bioengineering Institute (Nanjing Jiancheng Bioengineering Institute, Nanjing, China). Reagents for qPCR applications included SYBR Green for real-time PCR (TransGen Biotech, Beijing, China) and the RevertAid First Strand cDNA Synthesis Kit (CW0581, Beijing, China). For Western blotting analysis, RIPA buffer (high), which was used to extract the total protein, was purchased from Solarbio (Solarbio, Beijing, China), and antibodies against GRP78 (11587-1-AP), CHOP (15204-1-AP), BAX (50599-2-lg), and Bcl-2 (12789-1-AP) were purchased from Proteintech (Proteintech, Wuhan, China). Antibodies against ATF4 (1531R) and ATF6 (23094R) were purchased from Bioss (Bioss, Beijing, China). Antibody against caspase-3 (90437) was purchased from Abcam (Abcam, Cambridge, MA, USA) and antibody against GAPDH (AF7021) was purchased from Affinity Biosciences (Affinity, Cincinnati, OH, USA).

### 5.2. Cell Culture

The bovine mammary epithelial cell line MAC-T was kindly provided by Professor Hong Gu Lee (Konkuk University, Seoul, Korea). For in vitro analyses, MAC-T cells were maintained in high-glucose DMEM containing 10% FBS, 1% penicillin–streptomycin, 1 μg/mL hydrocortisone, and 5 μg/mL insulin and kept in a 37 °C incubator with 5% CO_2_.

### 5.3. Cell Treatment

Taraxasterol was dissolved in DMSO at a concentration of 50 mg/mL for storage and diluted to different concentrations in DMEM for cell treatments. The final concentration of DMSO in the treatment solutions prepared above was less than 0.1% (*v*/*v*). DON was dissolved in ethanol at a concentration of 10 mg/mL for storage and diluted to the specific concentrations needed. DON was first diluted to 0.05, 0.1, 0.2, 0.3, and 0.5 μg/mL for treatment to discover the toxicity of DON. Taraxasterol was diluted to 1, 5, 10, and 20 μg/mL for treatment.

### 5.4. Cell Viability Assay

Cell viability was measured using a CCK-8 kit (Dojindo Laboratories, Kumamoto, Japan) following the manufacturer’s instructions. MAC-T cells were seeded in 96-well plates at a density of 1 × 10^4^ cells/well. When the cells reached 70–80% confluence, they were treated with different concentrations of DON (0, 0.05, 0.1, 0.2, 0.3, 0.5 μg/mL) and taraxasterol (0, 1, 5, 10, and 20 μg/mL) for 24 h. A cell viability test was then performed, a total of 10 μL of CCK-8 reagent was added to each well, and the cells were incubated for 1.5 h at 37 °C. Then, the cell viability was measured using enzyme calibration. Cell viability was calculated via the absorbance value (OD) of each well measured at 450 nm. Subsequent experiments were conducted using the combination of 0.2 μg/mL DON and 10 μg/mL taraxasterol.

### 5.5. LDH Assay

LDH leakage was detected using an LDH kit (Nanjing Jiancheng Bioengineering Institute, Nanjing, China). Four groups (the control group, DON group, DON + taraxasterol group and taraxasterol group) of MAC-T cells were seeded in 6-well plates. DON and taraxasterol were administered when the cell confluence reached 70–80%. The cell culture medium was collected after 24 h and plated into 96-well plates with reagents from the LDH kit following the manufacturer’s instructions. LDH leakage was measured using enzyme calibration and calculated via the absorbance value (OD) of each well measured at 450 nm.

### 5.6. Measurement of Malondialdehyde (MDA), Glutathione (GSH), Total Antioxidant Capacity (T-AOC) and Total Superoxide Dismutase (T-SOD) Levels

The content of MDA and the activities of GSH, T-AOC, and T-SOD were detected using commercial kits. MAC-T cells were seeded in 6-well plates and simultaneously treated with DON and taraxasterol for 24 h when the cell confluence reached 70–80%. Then, the cells were collected to measure oxidation levels. The culture solution was also collected for later use. The cells were taken out with a cell scraper and transferred to a 1.5 mL centrifuge tube. Then, 500 µL of the extract was added, and the contents were mixed by homogenization. Subsequently, 100 µL of the mixture was transferred to another 1.5 mL centrifuge tube. The BCA kit determined the protein concentration. We measured the absorbance at 530 nm in the microplate reader. To determine the T-SOD activity, the cells were cultivated in a similar way, then the cell protein concentration was evaluated. The kit instructions were used to add the reagents, which were mixed well and kept at room temperature for 10 min, then the wavelength of 550 nm was colorimetrically detected. To determine the GSH level, cultured cells were taken out by cell scraping and transferred to a 1.5 mL centrifuge tube. A glass homogenizer was used for mixing; 100 µL of the precipitation solution was taken and centrifuged at 3500 rpm for 10 min, and then the supernatant was taken for detection. The absorbance was measured at 405 nm in the microplate reader.

### 5.7. Measurement of ROS Production

The intracellular ROS levels of MAC-T cells were measured with a DCFH-DA kit (Shanghai, China). MAC-T cells were seeded in 6-well plates and simultaneously treated with DON and taraxasterol for 24 h when the cell confluence reached 70–80%. The cells were then stained with 10 μL of DCFH-DA for 30 min at 37 °C in the dark. The cells were washed using 1× PBS to remove the unincorporated dye. The green fluorescence intensity was measured using the fluorescence microscope function of a Cytation five-cell imaging reader (BioTek Instruments, Winooski, VT, USA). Data were analyzed using Gen5 3.03 software (BioTek Instruments, Winooski, VT, USA).

### 5.8. Mitochondrial Membrane Potential Assay

The mitochondrial membrane potential assay uses the fluorescent dye JC-1 (Beyotime), which can be used for early detection of apoptosis. To conduct the assay, the MAC-T cells were seeded in a 6-well plate at a density of 2 × 10^5^/well. MAC-T cells were simultaneously treated with DON and taraxasterol for 24 h once the cell density reached approximately 80% confluence. Then the MAC-T cells were treated with methylbenzimidazole and the fluorescent probe dichloromethane iodide (JC-1) and incubated for 0.5 h. Observation and detection were performed using the fluorescence microscope functionality of a Cytation 5 imaging reader (BioTek Instruments).

### 5.9. RNA Isolation and Quantitative Real-Time Polymerase Chain Reaction (PCR)

Real-time PCR was conducted to measure the mRNA levels of glucose-regulated protein 78 kDa (GRP78), activating transcription factor 6 (ATF6) and the transcription factor C/EBP homologous protein (CHOP), genes in one of the important signaling pathways involved in ER stress. The β-actin, GRP78, ATF6, and CHOP coding sequences (CDS) were obtained from NCBI, and each CDS was input into online Primer 5 software, which was used to design upstream and downstream primer sequences (real-time PCR primer sequence list); the Primer Basic Local Alignment Search Tool (BLAST) at the National Center for Biotechnology Information (NCBI) was used for primer verification. Primer synthesis was completed by Suzhou Jinweizhi Company, and β-actin was used as the internal reference. MAC-T cells were seeded in 6-well plates and simultaneously treated with DON and taraxasterol for 24 h when the cell confluence reached 70–80%. Total RNA was isolated using TRIzol reagent (Invitrogen, Carlsbad, CA, USA). Complementary DNA (cDNA) was synthesized from 1 μg of RNA with the following procedure: 42 °C for 15 min and then 85 °C for 5 min. cDNA, SYBR, ddH2O, upstream primers, and downstream primers were mixed in eight-strip tubes and centrifuged briefly to ensure that the mixtures were at the bottom of the tubes. Then, the prepared eight-strip tube was put into the real-time fluorescent quantitative PCR instrument and subjected to the following protocol: melting at 94 °C for 30 s, predenaturation at 94 °C for 5 s, annealing at 60 °C for 15 s, and extension at 72 °C for 15 s. The number of cycles was generally 40 but was adjusted according to the use of different primers in the reaction. Quantitative fluorescence values were calculated using the 2^−ΔΔct^ method and used to calculate relative mRNA levels. The primers used for qPCR analyses are listed in Table 1.

### 5.10. Western Blotting

Western blotting was conducted to detect the relative protein levels of GRP78, ATF6, ATF4, and the transcription factor CHOP, which are involved in the ER stress signaling pathway, and caspase-3, B-cell lymphoma-2 (Bcl-2), and BCL2-associated X (Bax), which are involved in the cellular apoptosis signaling pathway. MAC-T cells were cultured in 6-well plates and simultaneously treated with DON and taraxasterol for 24 h when the cell confluence reached 70–80%. The total protein was extracted from the cells using RIPA buffer (high), and the concentration of the extracted protein was determined by the bicinchoninic acid (BCA) method. After the concentration of the extracted proteins was measured, each protein sample was mixed with loading buffer and heated at 100 °C for 5 min. Then, 10% sodium dodecyl sulfate polyacrylamide (SDS–PAGE) was carried out. Samples containing 20 μg of protein were added to each well of the SDS–PAGE gel and then transferred to a 0.45-µm ethyl acetate membrane with the half fry transfer method after electrophoresis. The membranes were washed with Tris-buffered saline with Tween (TBST) and incubated with the appropriate primary rabbit antibody (1:1000) specific for GRP78, ATF4, ATF6, CHOP, caspase-3, Bcl-2 or BAX at 4 °C overnight. After washing four times with TBST, the immunoblotted membranes were incubated with a horseradish peroxidase-labeled goat antirabbit immunoglobulin G (IgG) secondary antibody for 2 h at room temperature. Finally, using Pierce enhanced chemiluminescence (ECL) substrate, the protein bands were imaged on a chemiluminescence image analyzer.

### 5.11. Statistical Analyses

All experiments were performed independently at least three times, and the data are expressed as the mean ± standard error of the mean (SEM). GraphPad Prism software (Windows 5.02; GraphPad Software, Inc., San Diego, CA, USA) was used to test the significance of the data by one-way analysis of variance, and *p* < 0.05 was used to indicate a statistically significant difference.

## Figures and Tables

**Figure 1 toxins-14-00211-f001:**
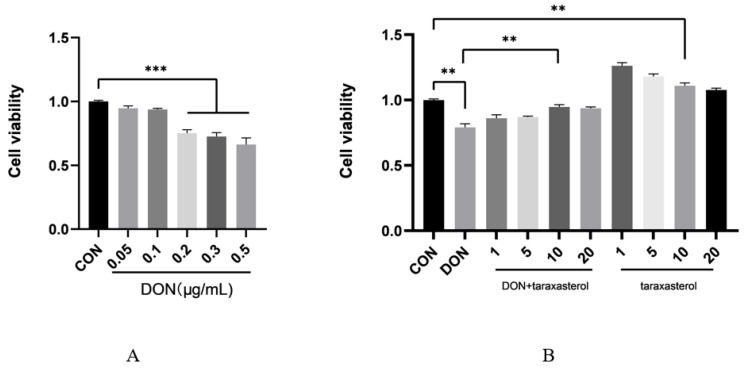
Effects of DON and taraxasterol on the viability of bovine mammary epithelial cells (MAC-T). (**A**) Viability of MAC-T cells treated with different concentrations (0.05, 0.1, 0.2, 0.3 and 0.5 μg/mL) of DON for 24 h. (**B**) Viability of MAC-T cells treated with 0.2 μg/mL DON, different concentrations (1, 5, 10 and 20 μg/mL) of taraxasterol or the combination of 0.2 μg/mL DON and different concentrations (1, 5, 10 and 20 μg/mL) of taraxasterol. All values are expressed as the mean ± SEM (*n* = 3). In the figure, ** *p* < 0.01, *** *p* < 0.001.

**Figure 2 toxins-14-00211-f002:**
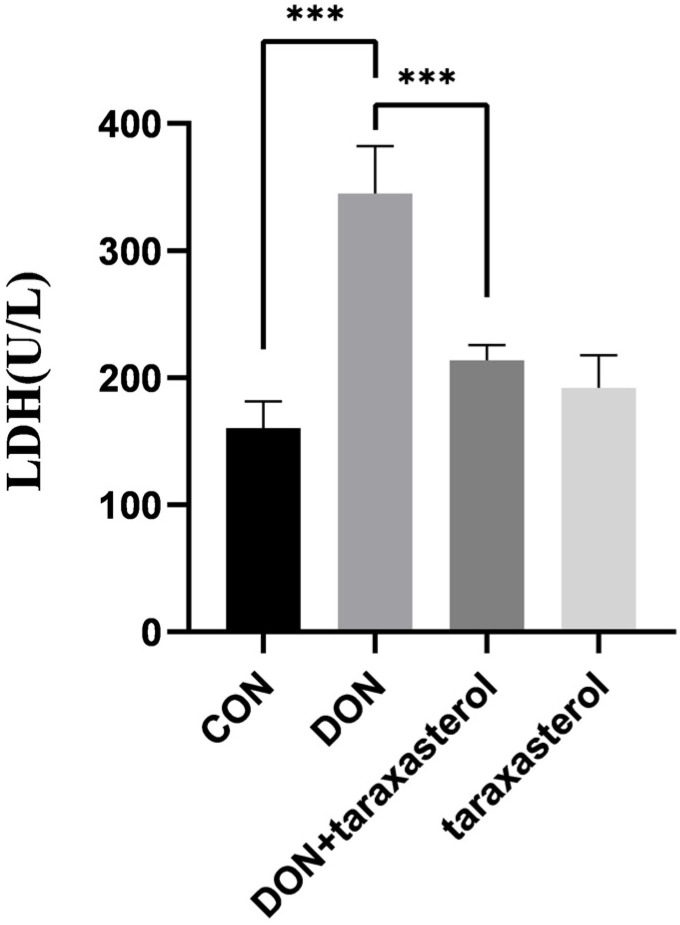
LDH leakage in cell culture medium after treatment with 0.2 μg/mL DON, 10 μg/mL taraxasterol or the combination of DON and taraxasterol for 24 h. All values are expressed as the mean ± SEM (*n* = 3). In the figure, *** *p* < 0.001.

**Figure 3 toxins-14-00211-f003:**
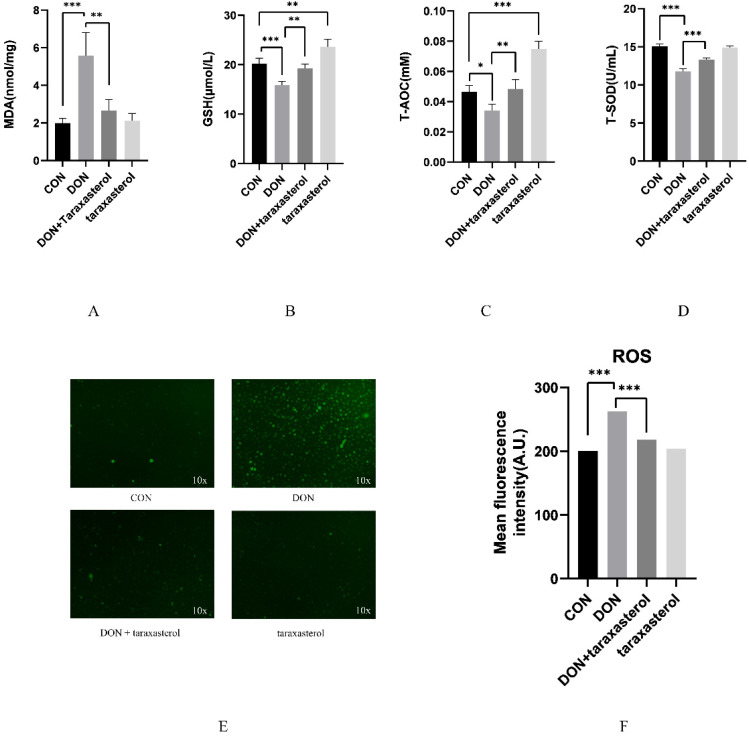
The antioxidant effects of taraxasterol against DON-induced cell damage. (**A**) MDA content in MAC-T cells. (**B**) GSH level in MAC-T cells. (**C**) T-AOC activity in MAC-T cells. (**D**) T-SOD activity in MAC-T cells. (**E**) After incubation with 5 μM 2′,7′-dichlorodihydrofluorescein diacetate (DCFH-DA), cells were washed and examined by fluorescence microscopy. Representative images from three independent experiments are shown. (**F**) ROS fluorescence intensity in MAC-T cells. All values are expressed as the mean ± SEM (*n* = 3). In the figure, * *p* < 0.05, ** *p* < 0.01, *** *p* < 0.001.

**Figure 4 toxins-14-00211-f004:**
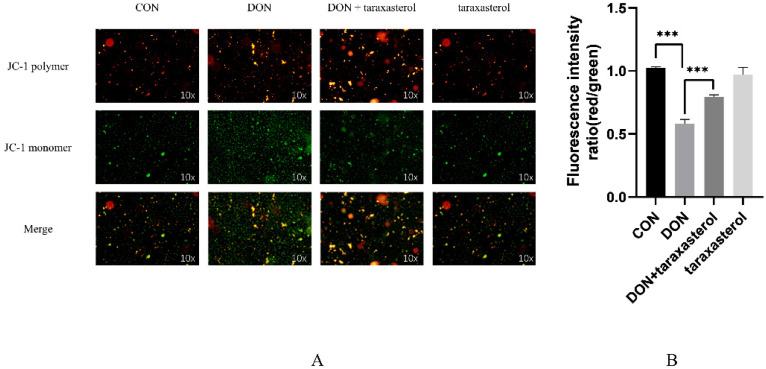
The mitochondrial membrane potential (ΔΨm, MMP) of MAC-T cells after treatment with 0.2 μg/mL DON and 10 μg/mL taraxasterol for 24 h. (**A**) Inverted fluorescence microscopy of MAC-T cells after JC-1 staining. Fluorescence analysis of MMP at 24 h using the probe JC-1. Representative images from three independent experiments are shown. (**B**) Fluorescence intensity ratio (indicating JC-1 polymer/JC-1 monomer) for each group. All values are expressed as the mean ± SEM (*n* = 3). In the figure, *** *p* < 0.001.

**Figure 5 toxins-14-00211-f005:**
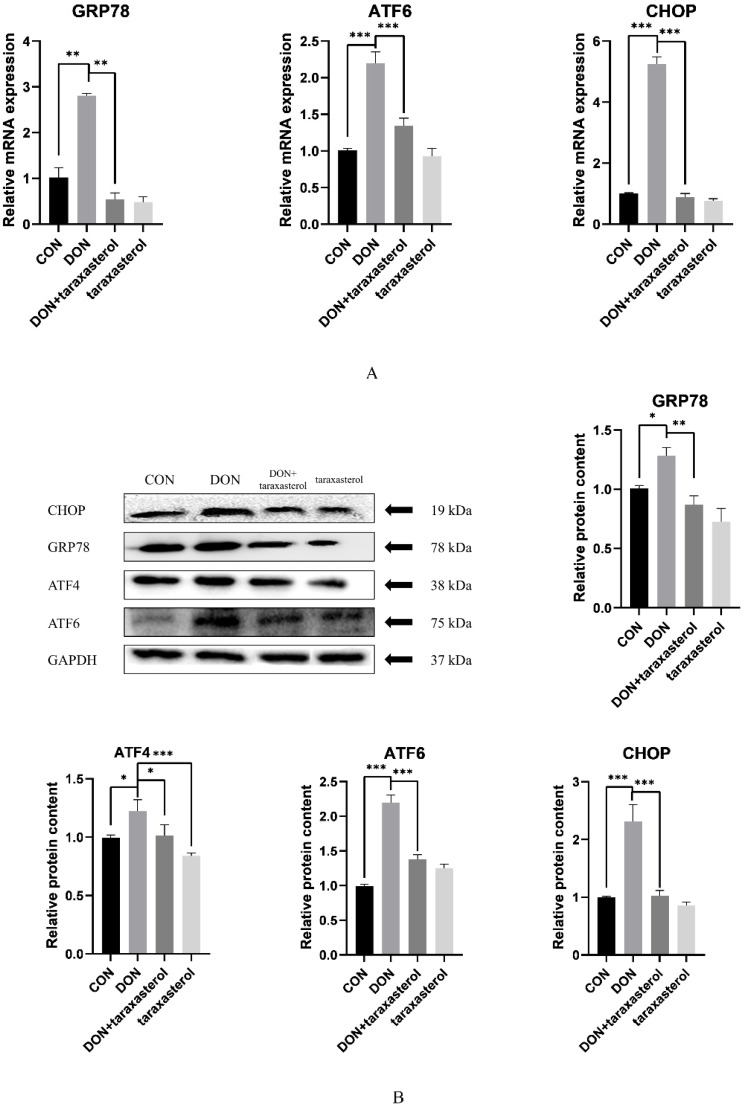
The protective effects of taraxasterol against ER stress induced by DON. (**A**) The relative gene expression of GRP78, ATF6 and CHOP in MAC-T cells after treatment with 0.2 μg/mL DON and 10 μg/mL taraxasterol for 24 h. (**B**) The relative protein levels of GRP78, ATF4, ATF6, and CHOP in MAC-T cells after treatment with 0.2 μg/mL DON and 10 μg/mL taraxasterol for 24 h. All values are expressed as the mean ± SEM (*n* = 3). In the figure, * *p* < 0.05, ** *p* < 0.01, *** *p* < 0.001.

**Figure 6 toxins-14-00211-f006:**
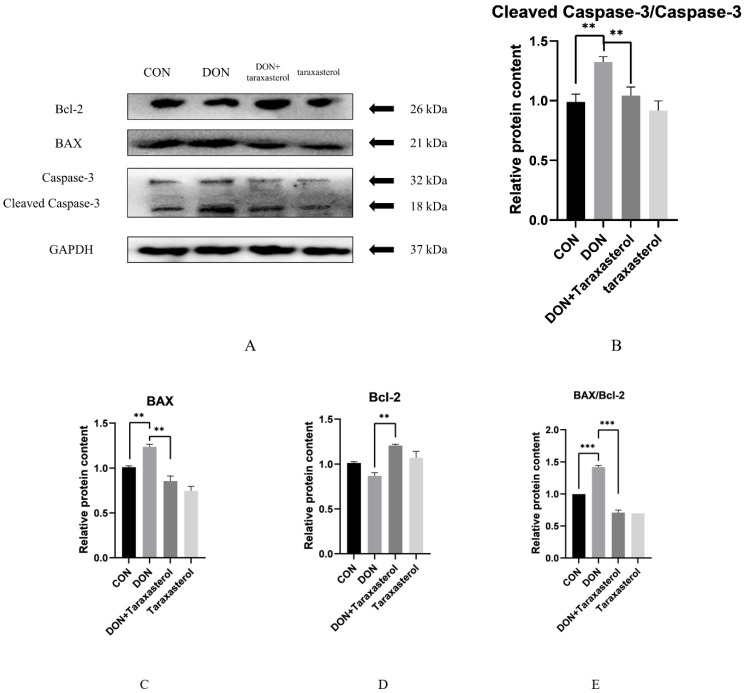
The protective effects of taraxasterol against DON-induced cell apoptosis. (**A**) Protein levels of Bcl-2, BAX and caspase-3. (**B**) Relative protein level of cleaved caspase-3/caspase-3. (**C**) Relative protein level of BAX. (**D**) Relative protein level of Bcl-2. (**E**) The ratio of the relative protein level of BAX to that of Bcl-2. All values are expressed as the mean ± SEM (*n* = 3). In the figure, ** *p* < 0.01, *** *p* < 0.001.

**Table 1 toxins-14-00211-t001:** Gene name and PCR primer sequences.

Gene	Forward Primer	Reverse Primer	GenBank Accession No.	Product Size (bp)
*β-actin*	5′-CCCTGGAGAAGAGCTACGAG-3′	5′-GTAGTTTCGTGAATGCCGCAG-3′	NM_173979.3	130
*GRP78*	5′-CGACCCCTGACGAAAGACAA-3′	5′-AGGTGTCAGGCGATTTTGGT-3′	NM_001075148.1	198
*ATF6*	5′-ATATTCCTCCGCCTCCCTGT-3′	5′-GTCCTTTCCACTTCGTGCCT-3′	XM_024989876.1	103
*CHOP*	5′-GAGCTGGAAGCCTGGTATGA-3′	5′-CTCCTTGTTTCCAGGGGGTG-3′	NM_001078163.1	90

## Data Availability

Data sharing not applicable.

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
