# Peer review of "Protective Effects of Taraxasterol against Deoxynivalenol-Induced Damage to Bovine Mammary Epithelial Cells"

_toxins, 2022, doi:10.3390/toxins14030211_

Round 1

Reviewer 1 Report

The authors presented a new submission of a previous paper. In this version, a blot of caspase 3, showing the precursor and the mature forms, has been provided in response to the reviewer's claim. The blot is not very convincing, and the image is quite dark. Can the authors provide another image of the caspase 3 blot?  (the authors have indicated n=3 experiments).

In the manuscript, there is other evidence related to the apoptosis activation by DON (MMP, BAX, BCL2), so that authors can suggest in the discussion that apoptosis mediators other than caspase 3 may be involved in the toxic effect exerted by DON.

Author Response

Response to Reviewer 1 Comments

Point 1: The authors presented a new submission of a previous paper. In this version, a blot of caspase 3, showing the precursor and the mature forms, has been provided in response to the reviewer's claim. The blot is not very convincing, and the image is quite dark. Can the authors provide another image of the caspase 3 blot?  (the authors have indicated n=3 experiments)

Response 1: Thank you for your valueble advise. We will provide a clearer image of caspase-3 this time.

Point 2: In the manuscript, there is other evidence related to the apoptosis activation by DON (MMP, BAX, BCL2), so that authors can suggest in the discussion that apoptosis mediators other than caspase 3 may be involved in the toxic effect exerted by DON.

Response 2: Thank you for your valueble advise. We will mention the apoptosis that DON induced may involve the changes of MMP, BAX, BCL2 as you recommended this time.

Reviewer 2 Report

The author corrected the manuscript as much as possible based on the reviewer’s comments.  

Author Response

The author corrected the manuscript as much as possible based on the reviewer’s comments.

Thank you again for reviewing our manuscript.

This manuscript is a resubmission of an earlier submission. The following is a list of the peer review reports and author responses from that submission.

Round 1

Reviewer 1 Report

In the article, the authors investigated the protective effects of taraxasterol against deoxynivalenol (DON)-induced cell damage in MAC-T bovine mammary epithelial cells. The results showed that taraxasterol significantly alleviated the decrease in cell viability, increase in LDH leakage and ROS levels, decrease in MMP, ER stress and apoptosis induced by DON. Data are linear and experiments are simple. The manuscript has been well organized and written. At this stage, the manuscript should be improved requires before acceptance.

Please find below my comments.

  1. Did the authors detect full-length caspase-3, not cleaved caspase-3? During apoptosis, full-length caspase-3 gets activated by proteolytic cleavage into the 17-19 kDa (p17, p18) and 12 kDa (p12) active subunits. The cleaved caspase-3 is an appropriate marker for apoptosis.
  2. Mitochondria membrane potential assay should be provided in Materials and Methods.
  3. Delete “*p < 0.05” in Fig 2.
  4. In 4.4 Cell viability assay, “1× 104 cells/well” should be corrected with superscript.

Author Response

Response to Reviewer 1 Comments

Point 1: Did the authors detect full-length caspase-3, not cleaved caspase-3? During apoptosis, full-length caspase-3 gets activated by proteolytic cleavage into the 17-19 kDa (p17, p18) and 12 kDa (p12) active subunits. The cleaved caspase-3 is an appropriate marker for apoptosis.

Response 1: Thank you for your valueble advise. We used the full-length caspase-3 as the marker for apoptosis in our research. We agree that cleaved caspase-3 is an appropriate marker for apoptosis. Caspase-3 (32kDa) is also a crucial part of the pathway of cell apoptosis. The article Emerging roles of caspase-3 in apoptosis. Porter AG, Jänicke RU. Cell Death Differ. 1999 Feb;6(2):99-104. doi: 10.1038/sj.cdd.4400476. PMID: 10200555. illustrated the the function of caspase-3 during the apoptosis. We believe the protein level of caspase-3 (32kDa) can be interfered by Bcl-2. So with the upregulation of the protein level of Bcl-2, the protein level of caspase-3 (32kDa) will be downregulated, the cell apoptosis will be reduced along with it. Similar results was shown in the article Curcumin Alleviates LPS-Induced Oxidative Stress, Inflammation and Apoptosis in Bovine Mammary Epithelial Cells via the NFE2L2 Signaling Pathway. Li R, Fang H, Shen J, Jin Y, Zhao Y, Wang R, Fu Y, Tian Y, Yu H, Zhang J. Toxins (Basel). 2021 Mar 12;13(3):208. doi: 10.3390/ toxins 13030208. PMID: 33809242; PMCID: PMC7999830. That’s why we decided to use the results of caspase-3 (32kDa) to illustrate our conclusion. We’re very happy to have further discussion with you concerning this process.

Point 2: Mitochondria membrane potential assay should be provided in Materials and Methods.

Response 2: Thank you for your valueble advise. The detailed operation steps concerning the evaluation of the mitochondrial membrane potential was supplemented.

Point 3: Delete “*p < 0.05” in Fig 2.

Response 3: Thank you for your kind reminding. The “*p < 0.05” in Fig 2. has been deleted as suggested.

Point 4: In 4.4 Cell viability assay, “1× 104 cells/well” should be corrected with superscript.

Response 4: Thank you for your kind remingding. The “1× 104 cells/well” has been corrected with superscript.

Reviewer 2 Report

The authors presented a study on the protective effects of taraxasterol against deoxynivalenol (DON)-induced damage to bovine mammary epithelial cells. Some data on the toxic effects of DON in the same cell model have already been published (doi: 10.1111/jpn.13180). Therefore, the novelties of the work only concern the protective action of taraxasterol. To explain the mechanism of action, the authors analyzed the involvement of ER stress. Data here presented indicate that taraxasterol alleviates the damage caused by DON mycotoxin through the attenuation of ER stress.

Most of the data shown in the manuscript support the authors' conclusions. One discrepancy relates to the effects of DON and taraxasterol on caspase 3, as noted in the comments.

Comments:

- The authors should indicate the code of the reagents and kits used in the experiments. Indeed, it is not always clearly deducible how the experiments were carried out. For example, paragraph 4.6 describes the experiments too superficially and does not allow the readers to understand how the experiments were performed.

- It is not clear whether the authors determined the activity of glutathione reductase, or if they determined the endogenous levels of reduced glutathione. The terms used by the authors are confusing. In paragraph 2.3 the authors indicate GSH (?) activity, while in the materials and methods they indicate the levels of GSH.

- In Figure 3E, it is necessary to stain the nuclei of the cells for normalization of the fluorescence correlated to the endogenous ROS

- In the materials and methods, the part concerning the evaluation of the mitochondrial membrane potential is not described-

 - It is not correct to state “mRNA expression levels”. The correct terms are “gene expression”, or

 “mRNA levels”.

- The data on the activation of caspase 3 does not seem to support the authors' conclusions. The authors state that treatment with DON determines the activation of caspase 3, as suggested by the increase in the levels of 35 KDa protein. However, caspase 3 is activated by proteolytic processing, which generate the active form of 17 KDa. From the WB images, the precursor form of caspase 3 can be observed, while the mature form is not evident. Since the caspase 3 level (35 KDa) decreases in samples with Taraxasterol, it would therefore appear that this compound activates the proteolysis of caspase 3. The authors should explain this discrepancy with what they claim in the text.

-There are several incorrect sentences, indicated with considerable superficiality, which authors must pay attention to and correct adequately throughout the text. For example “the decrease in capacity-AOC”; “decreases in SOD and GSH”; “the levels of MDA, GSH, T-SOD and T-AOC were tested” ; “by downregulating the expression of BAX and upregulating the expression of Bcl-2, the expression of caspase-3 was also suppressed” (this sentence is not correct), and more.

-The commercial source and codes of the antibodies used in the study are not reported.

- The names of the authors and their affiliations are not indicated in the manuscript. Other parts of the manuscript are missing.

-The authors are encouraged to show the original, uncropped blots, with the marker, in a supplementary figure.

Author Response

Response to Reviewer 2 Comments

Point 1: The authors should indicate the code of the reagents and kits used in the experiments. Indeed, it is not always clearly deducible how the experiments were carried out. For example, paragraph 4.6 describes the experiments too superficially and does not allow the readers to understand how the experiments were performed.

Response 1: Thank you for your valueble advise. We’ve made several changes according to your advise. The code of the reagents and kits were supplemented in the manuscript, and the detailed operation steps were added to make the readers understand the experiments better.

Point 2: It is not clear whether the authors determined the activity of glutathione reductase, or if they determined the endogenous levels of reduced glutathione. The terms used by the authors are confusing. In paragraph 2.3 the authors indicate GSH (?) activity, while in the materials and methods they indicate the levels of GSH.

Response 2: Thank you for your valueble advise. We’re sorry for the unclear expression. We’ve changed the expression “the levels of GSH” in the materials and methods to “the activity of GSH” as we mentioned in paragraph 2.3.

Point 3: In Figure 3E, it is necessary to stain the nuclei of the cells for normalization of the fluorescence correlated to the endogenous ROS

Response 3: Thank you for your valueble advise. The principle of our ROS kits was using DCFH-DA to stain the cells. We believe the current result of ROS is enough to show the changes of the MAC-T cells after treated with DON and taraxasterol respectively. There’re several articles using the similar strategy to detect the intracellular ROS level. For example, if you will be kind enough to go through the article named Curcumin Alleviates LPS-Induced Oxidative Stress, Inflammation and Apoptosis in Bovine Mammary Epithelial Cells via the NFE2L2 Signaling Pathway. Li R, Fang H, Shen J, Jin Y, Zhao Y, Wang R, Fu Y, Tian Y, Yu H, Zhang J. Toxins (Basel). 2021 Mar 12;13(3):208. doi: 10.3390/ toxins 13030208. PMID: 33809242; PMCID: PMC7999830., you will find a similar ROS detection. At the same time, we’re happy to provide the figures of the bright field to aviod unclarities.

Point 4: In the materials and methods, the part concerning the evaluation of the mitochondrial membrane potential is not described-

Response 4: Thank you for your valueble advise. The detailed operation steps concerning the evaluation of the mitochondrial membrane potential was supplemented.

Point 5: It is not correct to state “mRNA expression levels”. The correct terms are “gene expression”, or “mRNA levels”.

Response 5: Thank you for your valueble advise. The correction of the wrong expression have been made.

Point 6: The data on the activation of caspase 3 does not seem to support the authors' conclusions. The authors state that treatment with DON determines the activation of caspase 3, as suggested by the increase in the levels of 35 KDa protein. However, caspase 3 is activated by proteolytic processing, which generate the active form of 17 KDa. From the WB images, the precursor form of caspase 3 can be observed, while the mature form is not evident. Since the caspase 3 level (35 KDa) decreases in samples with Taraxasterol, it would therefore appear that this compound activates the proteolysis of caspase 3. The authors should explain this discrepancy with what they claim in the text.

Response 6: Thank you for your valueble advise. Caspase-3 (32kDa) is a crucial part of the pathway of cell apoptosis. The article Emerging roles of caspase-3 in apoptosis Porter AG, Jänicke RU. Cell Death Differ. 1999 Feb;6(2):99-104. doi: 10.1038/sj.cdd.4400476. PMID: 10200555. illustrated the the function of caspase-3 during the apoptosis. We believe the protein level of caspase-3 (32kDa) can be interfered by Bcl-2. So with the upregulation of the protein level of Bcl-2, the protein level of caspase-3 (32kDa) will be downregulated, the cell apoptosis will be reduced along with it. Similar results was shown in the article Curcumin Alleviates LPS-Induced Oxidative Stress, Inflammation and Apoptosis in Bovine Mammary Epithelial Cells via the NFE2L2 Signaling Pathway. Li R, Fang H, Shen J, Jin Y, Zhao Y, Wang R, Fu Y, Tian Y, Yu H, Zhang J. Toxins (Basel). 2021 Mar 12;13(3):208. doi: 10.3390/ toxins 13030208. PMID: 33809242; PMCID: PMC7999830.That’s why we decided to use the results of caspase-3 (32kDa) to illustrate our conclusion. We’re very happy to have further discussion with you concerning this process.

Point 7: There are several incorrect sentences, indicated with considerable superficiality, which authors must pay attention to and correct adequately throughout the text. For example “the decrease in capacity-AOC”; “decreases in SOD and GSH”; “the levels of MDA, GSH, T-SOD and T-AOC were tested” ; “by downregulating the expression of BAX and upregulating the expression of Bcl-2, the expression of caspase-3 was also suppressed” (this sentence is not correct), and more.

Response 7: Thank you for your valueble advise. The incorrect sentences you kindly mentioned have been corrected.

Point 8: The commercial source and codes of the antibodies used in the study are not reported.

Response 8: We’re very sorry for our negligence. The commercial source and codes of the antibodies used in the study have been supplemented.

Point 9: The names of the authors and their affiliations are not indicated in the manuscript. Other parts of the manuscript are missing.

Response 9: Thank you for your kind reminding. We’re not quite sure if there’s any misunderstanding, the names of the authors and their affiliations were indicated in the manuscript below the title of the article.

Point 10: The authors are encouraged to show the original, uncropped blots, with the marker, in a supplementary figure.

Response 10: We’re happy to show the original, uncropped blots. Please see the supplementary figure.

Round 2

Reviewer 2 Report

The authors submitted a revised version in light of the comments raised in the first revision.
Although a good part of the points raised was resolved by the authors, some critical aspects still remain unresolved.
In the previous review, the authors were asked for uncropped images of the blots. Unfortunately, these images continue to be cropped, so their evaluation by the reviewer remains elusive. This aspect concerns, once again, the interpretation of the mechanism of caspase 3 activation. As the authors know, activation of pro-caspase 3 (37 KDa) generates the active cleaved form. This activation is not shown in the western blot, where only the inactive pro-caspase 3 (37 KDa) is evidenced.
Otherwise, the absence of apoptosis in control untreated cells, where pro-caspase 3 levels are just 20% lower than in DON treated cells, would be difficult to interpret. In order to support the activation of the apoptotic process, experimental evidence is needed to support the hypothesis (DNA laddering; Annexin V; etc). No such evidence has been presented in the manuscript. 

The authors continue to confuse the determination of reduced glutathione (GSH) levels with the enzymatic activity of glutathione reductase. The manuscript reports data on glutathione (GSH) levels, not the enzymatic activity of glutathione reductase. This error should be adequately corrected throughout the manuscript.